# The Roles of Plant Hormones and Their Interactions with Regulatory Genes in Determining Meristem Activity

**DOI:** 10.3390/ijms20164065

**Published:** 2019-08-20

**Authors:** Ze Hong Lee, Takeshi Hirakawa, Nobutoshi Yamaguchi, Toshiro Ito

**Affiliations:** 1Division of Biological Science, Graduate School of Science and Technology, Nara Institute of Science and Technology, 8916-5, Takayama, Ikoma, Nara 630-0192, Japan; 2Precursory Research for Embryonic Science and Technology, Japan Science and Technology Agency, 4-1-8, Honcho, Kawaguchi-shi, Saitama 332-0012, Japan

**Keywords:** *Arabidopsis thaliana*, shoot apical meristem, floral meristem, auxin, cytokinin, WUSCHEL, CLAVATA, AGAMOUS

## Abstract

Plants, unlike animals, have developed a unique system in which they continue to form organs throughout their entire life cycle, even after embryonic development. This is possible because plants possess a small group of pluripotent stem cells in their meristems. The shoot apical meristem (SAM) plays a key role in forming all of the aerial structures of plants, including floral meristems (FMs). The FMs subsequently give rise to the floral organs containing reproductive structures. Studies in the past few decades have revealed the importance of transcription factors and secreted peptides in meristem activity using the model plant *Arabidopsis thaliana*. Recent advances in genomic, transcriptomic, imaging, and modeling technologies have allowed us to explore the interplay between transcription factors, secreted peptides, and plant hormones. Two different classes of plant hormones, cytokinins and auxins, and their interaction are particularly important for controlling SAM and FM development. This review focuses on the current issues surrounding the crosstalk between the hormonal and genetic regulatory network during meristem self-renewal and organogenesis.

## 1. Introduction

Throughout their entire life cycle, plants possess a small group of pluripotent stem cells in their meristems [1,2,3]. The shoot apical meristem (SAM) at the top of the plant is responsible for postembryonic growth and gives rise to plant aerial structures (Figure 1a) [4]. To sustain proper continuous growth, the SAM maintains the balance between self-renewal of stem cells and cell recruitment for lateral organ formation. Stem cells in the SAM produce daughter cells, which remain stem cells or become differentiated. The SAM also establishes phyllotaxy, i.e., the arrangement of lateral organs along the stem [5,6,7,8,9]. (Figure 1b). In *Arabidopsis thaliana*, the SAM produces rosette leaves during the vegetative phase (Figure 1a). Just after the floral transition, a few cauline leaves and branches form from the SAM [10] (Figure 1a). During the reproductive phase, flowers are produced from the SAM (Figure 1a). The phase-specific activities of the SAM determine plant architecture (Figure 1a). The SAM is divided into the outer tunica and corpus. The outer layer is composed of epidermal (L1) and subepidermal (L2) layers (Figure 1c). L1 and L2 are single cell layers that divide anticlinally to the plane of the tissue surface. By contrast, the corpus, or inner L3, is a collection of cells whose division occurs in all planes [11]. The SAM harbors a set of stem cells within the central zone (CZ) surrounded by the peripheral zone (PZ) (Figure 1c). Cells located in the CZ divide slowly, while cells located in the PZ divide rapidly (Figure 1c). The balance of cell division in the CZ and PZ determines organ size and number. The PZ is active in the production of lateral organs. The rib zone (RZ) provides multipotent cells to form stem cells, which support the SAM. The CZ acts as a reservoir of stem cells. CZ activity is maintained by the underlying organizing center (OC). The CZ and OC partially overlap and form the stem cell niche (Figure 1c) [12,13].

The WUSCHEL (WUS) transcription factor and CLAVATA (CLV) ligand–receptor system are key determinants of meristematic activity in the SAM [14,15,16,17,18]. Mutations in the *WUS* gene result in premature termination of the SAM after a few organs have formed [16]. *WUS*, which is expressed in the OC, controls biological processes through the transcriptional regulation of downstream target genes related to meristem growth, cell division, and hormonal signaling [19,20,21,22]. In particular, WUS specifies stem cell identity, partially through the direct activation of *CLV3*. On the other hand, the loss-of-function *CLV3* mutant aberrantly accumulates stem cells in the SAM [23]. *CLV3*, which is expressed in the CZ, encodes a founding member of the CLAVATA3/ESR-RELATED (CLE) family of small peptides. The CLV3 peptide binds to the CLV receptor proteins [24]. *CLV1* encodes a receptor kinase containing an extracellular leucine-rich repeat domain and an intracellular kinase domain. CLV2 is similar to CLV1 but lacks a cytoplasmic kinase domain [12,25,26]. CLV1 forms homodimers, while CLV2 forms heterodimers with the receptor-like cytoplasmic kinase CORYNE (also known as SUPPRESSOR OF LLP1 2 (SOL2)) for signal transduction [12,27,28,29,30]. CLV signaling itself represses *WUS* expression to restrict its spatial expression domain [12,13]. The negative feedback loop between stem cells and the OC mediated by these two proteins ensures stem cell homeostasis in the SAM and indefinite organ formation (Figure 1d). Components of the CLV-WUS negative feedback pathway are well conserved in model and crop plants [31,32]. In rice (*Oryza sativa*), *FON2-LIKE CLE PROTEIN1* (*FCP1*) and *FCP2*, which encode CLV3-related CLE proteins, play key roles in vegetative SAM maintenance [33]. FCP1 and FCP2 act redundantly to repress *WUSCHEL RELATED HOMEOBOX4* (*WOX4*) expression [33]. In maize (*Zea mays*), ZmFCP1 also acts as a ligand for the receptor kinase FASCIATED EAR3 (FEA3) [34,35]. ZmFCP1 and FEA3 negatively regulate *ZmWUS1* expression to maintain reproductive SAM activity [34,35]. Thus, molecular evidence indicates that the CLV-WUS pathway is critical for SAM maintenance in higher plants.

During the reproductive phase, SAMs give rise to floral meristems (FMs) in a regular, geometric fashion. FMs also contain stem cell populations during their early growth stages. The balance between the rates of stem cell proliferation and differentiation in the FM is pivotal for the proper formation of flower organs, such as sepals, petals, stamens, and carpels (Figure 1d) [36]. Floral organ patterning is determined by the combined actions of homeotic genes [37,38]. Sepals, stamens, carpels, and petals become visible in order (Figure 1d–f). Like the SAM, FM activity is also regulated by WUS activity during floral organ formation [16]. Mutations in *WUS* also result in premature termination of the FM. The regulatory principles that determine stem cell homeostasis in the SAM and FM are largely similar to each other. Like the SAM, the FM is also maintained by the CLV–WUS pathway in higher plants. In rice, *FLORAL ORGAN NUMBER1* (*FON1*) and *FON2* control FM sizes [39,40,41]. *FON1* encodes an ortholog of the CLV1 receptor, whereas *FON2* encodes a CLV3-like protein. These two proteins act in the same pathway for FM maintenance [39,40]. In tomato, the FACIATED AND BRANCHED (FAB) receptor and the SlCLV3 (ligand) signaling pathway also control FM size [42]. In Arabidopsis, the SAM is indeterminate and the FM is determinate, while SAMs can also be determinate in many species and can vary within a single plant species [6,43,44,45]. Precocious FM termination leads to the formation of fewer floral organs. By contrast, delayed FM termination leads to increased numbers of floral organs. Thus, the timing of FM termination is crucial for the production of a fixed number of organs, including the female structure, the gynoecium. This termination is controlled by multiple gene regulatory networks. The precise control of the termination of the FM by multiple factors ensures stem cell homeostasis in the FM and the formation of determinate floral organs.

Plant hormones (also known as phytohormones) are signaling molecules that influence a variety of physiological processes. Phytohormones are classified into several different groups depending on their chemical structures [46,47]. Plant hormones are synthesized in one location and move to other locations in the plant. Hormones trigger many biological and cellular processes in locally targeted cells, such as seed dormancy, growth, metabolism, organ formation, reproduction, and stress responses [48,49,50,51,52,53]. Plant hormone biosynthesis, transport, perception, signal transduction, and downstream effects coordinate the hormonal control of cell division, growth, and differentiation.

Recent studies of the regulators of the SAM and FM have revealed that plant hormones contribute to the fine-tuning of meristem maintenance and organ formation. Cytokinin and auxin are two core plant hormones that function in the SAM and FM regulatory network. Cytokinins are required for cell division in meristematic tissues, whereas auxins promote organ formation, growth, and differentiation. Cytokinins and auxins often work together in multiple organs, tissues, and cells [46]. Although these two hormones were originally considered to be antagonists, recent studies have revealed their synergistic interactions as well [54,55,56]. In this review, we discuss recent findings on the molecular basis of SAM and FM regulation, with an emphasis on the roles of cytokinin and auxin.

## 2. Hormonal Control of Meristematic and Primordial Fate in the SAM

### 2.1. The Role of Cytolonin in Specifying Meristem Fate in the SAM

#### 2.1.1. Cytokinin and Its Role in Specifying the Fate of the SAM

Cytokinin, which triggers cell division, was first discovered more than 50 years ago. This plant hormone controls every aspect of plant growth and development, including meristem function, vascular development, stress responses, and senescence. Cytokinin maintains the stem cell population in the SAM. In Arabidopsis, high levels of cytokinin signaling induce ectopic *WUS* expression and lead to stem cell fate in the surrounding cells (Figure 2a). This finding suggests that cytokinin is sufficient for the induction of *WUS* and the specification of stem cell fate. *WUS* levels are regulated by two different cytokinin-dependent pathways: CLV-dependent and CLV-independent pathways [58]. Cytokinins are transported from root to shoot and accumulate in the OC of the SAM [59]. Cytokinin activity is further fine-tuned by cytokinin biosynthesis, degradation, transport, and signaling [60,61]. Tissue-specific transcription factors and environmental signals also contribute to this activity [62].

#### 2.1.2. Cytokinin Biosynthesis Controls WUS Expression in the SAM

Cytokinin biosynthesis in Arabidopsis begins with the addition of a prenyl group to the N6 position of adenosine diphosphate/adenosine triphosphate (ADP/ATP). This reaction is catalyzed by adenosine phosphate-isopentenyltransferase (IPT) proteins [63]. Arabidopsis contains 11 IPT homologs [64,65], which are differentially expressed to produce cytokinins in a tissue-specific manner. In the SAM, *IPT7* expression is activated by another pluripotency factor, SHOOT MERISTEMLESS (STM) (Figure 2a) [66]. IPT7 is localized to the mitochondria [63,67]. The resulting N6-isopentenyladenine (iP) ribotides produced via IPTs are subsequently converted to trans-zeatin (tZ—the most abundant form of cytokinin in plants) via hydroxylation of the isoprenoid side chain [68,69]. Another group of enzymes, LONELY GUY (LOG) cytokinin nucleoside 5′-monophosphate phosphoribohydrolases, contribute to the production of cytokinin [70,71]. *LOG* genes were originally identified in rice [70]. Arabidopsis contains nine *LOG* genes (*LOG1* to *LOG9*) [71]. Among these, *LOG4* and *LOG7* are expressed in the L1 layer of the SAM and floral primordia, respectively [72]. The cytokinin pathway also contributes to *WUS* expression in response to environmental conditions, such as changes in light and nitrate availability (Figure 2a) [73,74,75]. CYTOKININ OXIDASE5 (CKX5) and CKX6 catalyze the irreversible degradation of cytokinin. CKX5 and CKX6 repress *WUS* expression via the degradation of cytokinin in the dark [74]. Cytokinins also mediate stem cell size through *WUS* expression in response to nutritional status. Grafting experiments revealed that cytokinin precursors function as long-distance signals to control SAM size and *WUS* expression [75]. Cytokinin accumulation is adjusted accordingly based on these environmental inputs. The resulting cytokinin functions as a systemic signal to the SAM and determines *WUS* expression and stem cell activity. Thus, cytokinin biosynthesis is mediated by both multiple enzymes and environmental inputs and governs *WUS* expression in the SAM.

#### 2.1.3. Cytokinin Transport and its Role in Specifying the Fate of the SAM

Cytokinin transport is mediated by three types of cytokinin transporters: purine permeases (PUPs), equilibrative nucleoside transporters (ENTs), and ATP-binding cassette G (ABCG) transporters [76,77,78,79,80,81]. Arabidopsis contains 23 PUP and 8 ENT family members [76,77]. PUP1, PUP2, PUP14, and ENT1 control cytokinin uptake, as revealed in a yeast expression system [76,82,83]. On the other hand, the ABCG14 transporter coordinates cytokinin export [80,81,84,85,86]. The *ABCG* transporter gene family, comprising 28 genes, plays diverse roles in cytokinin transport [86,87]. Although *ABCG14* is mainly expressed in roots, a mutation in *ABCG14* leads to reduced shoot growth [80,81]. ABCG14 localizes to the plasma membrane and transports tZ, as revealed by a tracer experiment [81]. Phenotypic and molecular experiments suggest that long-distance cytokinin transport is essential for the regulation of SAM activity. Whether these transporters contribute to the establishment of cytokinin activity at the OC of the SAM has not yet been addressed (Figure 2a).

#### 2.1.4. Another Feedback Loop between Cytokinin Signal Transduction and WUS Specifies SAM Fate

Cytokinin signaling is mediated by three ARABIDOPSIS HISTIDINE KINASE (AHK) receptors, AHK2, AHK3, and AHK4 (also known as CYTOKININ RESPONSE1), through a multistep His–Asp phosphorelay similar to that found in bacterial two-component signaling systems. AHK receptors contain both histidine kinase and receiver domains. Upon sensing cytokinin, the AHKs autophosphorylate at a His residue and transfer the phosphate to an aspartate residue on the receiver domain. Subsequently, this phosphate is transferred to a histidine residue on authentic histidine-containing phospho-transmitters (AHPs) [88,89,90,91,92]. This phosphorylation step allows the AHPs to be translocated from the cytoplasm to the nucleus to activate Arabidopsis response regulators (ARRs). To date, 22 ARR genes have been identified. Typical ARRs are categorized into type A or type B [93,94]. Type-B ARRs are transcriptional activators that promote the cytokinin response, whereas most type-A ARRs are transcriptional repressors [89,95]. Type-B ARRs, such as ARR1, ARR10, and ARR12, activate *WUS* expression by directly binding to its promoter [96,97,98,99,100]. The type-B ARR binding cis-element located in the promoter region of *WUS* plays primary roles in the activation of this gene by ARR1, ARR10, and ARR12 (Figure 2a) [98]. WUS represses the expression of type-A ARR genes *ARR5*, *ARR6*, *ARR7*, and *ARR15* (Figure 2a) [101]. WUS binds directly to the *ARR7* and *ARR15* promoters. Thus, WUS possesses another feedback loop controlling its expression pattern in the SAM.

#### 2.1.5. WUS and HEC Compete for Shared Cytokinin Signal Transduction Genes to Specify SAM Fate

The interplay between WUS–CLV and cytokinin is fine-tuned by tissue-specific transcription factors. The basic helix-loop-helix (bHLH) transcription factor HECATE1 (HEC1) is directly repressed by high concentrations of WUS protein (Figure 2a) [101,102,103,104]. HEC1 physically interacts with other bHLH transcription factors, such as HEC2 and HEC3. Consistent with this role in repression, HEC1 is expressed throughout the SAM except in the OC, where WUS highly accumulates. Ectopic expression of *HEC1* in the OC interferes with the maintenance of the SAM. HEC1 represses *CLV3* expression and activates type-A *ARR7* and *ARR15* expression (Figure 2a). Shared target genes of HEC1 and WUS, such as *ARR7* and *ARR15*, are regulated in an opposite manner. How this competitive regulation by HEC1 and WUS is regulated remains to be elucidated. Multiple feedback systems mediated by hormonal components and transcription factors act in parallel to control meristematic fate and *WUS* expression in the SAM. A feedback circuit-driven regulatory mechanism is a common strategy for reliable, irreversible cell fate determination [105,106,107,108].

### 2.2. The Role of Auxin in Specifying Meristem and Primordium Fate in the SAM

#### 2.2.1. AUXIN RESPONSE FACTORs Specify Meristem and Primordium Fate in the SAM

Auxin controls almost every aspect of plant growth and development, including cell division, elongation, and differentiation, with important effects on the final shapes and functions of plant cells and tissues. During SAM development, a classic role of auxin is to specify organ primordium fate in the PZ of the SAM. Mutants in components of auxin biosynthesis, transport, and signaling exhibit naked inflorescences lacking flowers, even though the earlier products from the SAM, such as rosette leaves, are generally present (Figure 2a) [109,110,111,112]. Recent evidence suggests that in addition to specifying organ primordium fate, auxin specifies meristematic fate in the SAM (Figure 2a). AUXIN RESPONSE FACTORs (ARFs) are transcription factors that play roles in specifying these identities [113,114]. To date, 23 *ARF* genes have been identified in Arabidopsis. ARFs function as transcription factors by binding to auxin-responsive elements (AuxREs) in the promoters of their target genes [113,114]. Individual ARFs control distinct developmental processes. Recent studies have identified downstream genes of ARFs, which play key roles in SAM development.

#### 2.2.2. MP/ARF5 Specifies Meristem and Primordium Fate in the SAM

Among ARF transcription factors, ARF5 (also known as MONOPTEROS (MP)) plays a key role in specifying meristematic and primordium fate by orchestrating gene expression [114,115,116,117,118]. MP is a canonical ARF (class A) that modulates auxin signaling [119,120]. MP dimerizes with AUXIN/INDOLE-3-ACETIC ACID (Aux/IAA) repressor proteins in the absence of auxin [115,116,117,118]. In the nucleus, auxin is perceived by TRANSPORT INHIBITOR RESISTANT1 (TIR1)/AUXIN SIGNALING F-BOX proteins [121,122]. These F-box proteins form the substrate recognition subunits of SKP-CULLIN-box (SCF) ubiquitin ligases [123]. Upon sensing auxin, Aux/IAA proteins are degraded through the action of SCF^TIR1^ ubiquitin ligase. MP activity is subsequently derepressed and triggers the expression of many auxin response genes. MP proteins are present at low levels in the CZ and at high levels in the PZ of the SAM [124,125]. Consistent with this accumulation pattern, MP modulates different downstream target genes in the CZ and PZ. To control meristematic fate in the CZ, MP directly represses *ARR7/ARR15* and activates *AHP6* through the regulation of cytokinin homeostasis (Figure 2a) [91,126]. AHP6 establishes inhibitory fields of cytokinin signaling to define organ initiation sites [91]. ARR7 and ARR15 integrate cytokinin and auxin signals and relay them to the WUS–CLV network [126]. MP also represses the expression of *DORNRÖSCHEN*/*ENHANCER OF SHOOT REGENERATION1* (*DRN*/*ESR1*) to promote *CLV3* expression (Figure 2a) [116,117,118,127,128]. Whether other transcription factors are required for the transcriptional activation of *CLV3* by DRN at the CZ of the SAM has not yet been addressed.

To specify primordium fate in the PZ, MP directly activates the auxin transporter gene *PIN-FORMED1* (*PIN1*), the floral meristem identity gene *LEAFY* (*LFY*), the organ size regulatory genes *AINTEGUMENTA* (*ANT*) and *AINTEGUMENTA-LIKE6* (*AIL6*), and the abaxial identity gene *FILAMENTOUS FLOWER* (*FIL*) [129,130,131,132,133,134,135]. The MP homodimer might bind to the *LFY* promoter via an AuxRE variant for MP homodimer binding [136]. MP also induces the expression of *MACCHI-BOU4* (*MAB4*) family genes in the PZ to control basipetal auxin transport (Figure 2a) [137]. The target specificity of MP occurs in a zone-dependent manner, perhaps due to threshold levels of MP protein, different affinities of MP binding sites, and/or chromatin-mediated regulation of gene expression. In fact, the expression of several MP targets, which specify primordium identity, requires the activity of Switch/Sucrose Non-Fermentable (SWI/SNF) family chromatin remodelers [138,139,140,141]. In the presence of auxin, MP interacts with SWI/SNF chromatin remodelers to open up the promoter regions of downstream target genes for their activation [141]. This structural change in chromatin allows additional transcription factors and/or general transcriptional machinery (Figure 2b). Hence, MP coordinates fate specification in the meristem and primordium in response to auxin in the SAM. The crosstalk between hormonal transcription factors and epigenetic regulators plays prominent roles in fate specification.

#### 2.2.3. ETT/ARF3 and ARF4 Switch Off Meristematic Cell Fate in the PZ of the SAM

ARF3 (also known as ETTIN (ETT)) and ARF4 are noncanonical class B ARFs that lack a domain for interaction with Aux/IAA proteins [113,142,143]. Both ETT and ARF4 are highly expressed in the PZ of the SAM [144]. ETT shares redundant functions with ARF4 during plant development. Unlike MP, an auxin-dependent interaction between ETT and process-specific transcription factors determines the transcriptional activity of ETT [144]. The molecular nature of the interaction between ETT and auxin remains to be addressed. To specify primordium fate, an ETT-FIL dimer directly represses the expression of the pluripotency factor gene *STM* through histone deacetylation and negatively regulates the STM target, *IPT7* (Figure 2a,c) [145]. Since *FIL* is a direct target of MP, MP also indirectly helps specify primordium fate by terminating the meristem (Figure 2a) [145]. The organogenic program that terminates the meristem is conserved between plants and animals. Furthermore, genome-wide analyses identified direct targets of ETT whose expression is ETT- and auxin-dependent. ETT directly controls the expression of *LFY* and the auxin biosynthetic gene *YUCCA4* (*YUC4*) in an auxin-dependent manner (Figure 2a) [144,146,147]. Like MP, ETT might also specify primordium fate via *LFY* in the PZ [144]. Furthermore, ETT might provide direct positive feedback regulation of auxin biosynthesis by activating *YUC4* during primordium fate specification. The proper control of gene activation and repression by auxin-dependent transcription factors switches the fates of meristematic and primordium cells.

## 3. Hormonal Control of the Termination of Meristematic Fate in the FM for Subsequent Organogenesis

### 3.1. The Roles of Cytokinin and Auxin in FM Development and Subsequent Organ Differentiation

Unlike the SAM, the FM is determinate, and its meristematic fate must be terminated. WUS ensures the maintenance of the stem cell pool in the SAM and FM. The MADS-box transcription factor AGAMOUS (AG) plays central roles in early stages of FM development and the termination of the meristematic fate of the FM during later stages of development [36,45,148,149,150,151]. In addition to mutations in *AG*, ectopic *WUS* expression beyond stage 6 is sufficient to trigger indeterminacy of the FM. AG is expressed throughout FM formation beginning at stage 3 of flower development and terminates FM activity at stage 6 of flower development by regulating gene expression. Recent studies have shown that AG influences cytokinin and auxin homeostasis during both the earlier and later stages of FM development. After FM termination, cytokinin and auxin homeostasis play key roles in gynoecium formation.

### 3.2. AG Controls Auxin and Cytokinin Levels during FM Formation

During the early stages of FM formation, AG controls FM activity by maintaining proper hormone levels. AG modulates *ETT* expression partially though *GIANT KILLER* (*GIK*), which encodes an AT-HOOK MOTIF CONTAINING NUCLEAR LOCALIZED (AHL) protein [152,153]. AHL family proteins interact with each other to form homo- or heterodimer complexes. AHLs also interact with other components, such as DOMAIN OF UNKNOWN FUNCTION296 (DUF296) [154]. The availability of many different forms of AHL protein complexes might enable the fine-tuning of downstream genes of AG in a spatiotemporal manner. ETT then directly represses the expression cytokinin biosynthetic genes *IPT3*, *IPT5*, and *IPT7* and mediates cell-cycle-related gene expression [155]. AG also activates another downstream target, *SUPERMAN* (*SUP*), which encodes a transcriptional repressor with a C2H2 zinc-finger DNA-binding domain and an EAR repression motif [156,157,158,159]. *SUP* downregulates the expression of *YUC1* and *YUC4* to decrease local auxin biosynthesis. SUP interacts with histone modification enzymes to deposit negative histone marks, thereby silencing gene expression [160,161,162]. Proper hormone levels are required for maintaining FM activity and floral organ formation [162].

### 3.3. Proper Control of Cytokinin Homeostasis Promotes Cell Proliferation in the Terminating FM

During FM termination, AG controls cytokinin homeostasis [155]. Exogenous cytokinin treatment enhances the FM indeterminacy of *ag* mutants [155]. This phenotype is enhanced in cytokinin-treated *ag ett* double mutants, suggesting that ETT genetically interacts with cytokinin to regulate FM determinacy. ETT integrates AG and auxin inputs to repress cytokinin activity. ETT negatively controls cytokinin biosynthesis. Similar to its activity during FM formation, ETT directly represses the expression of the cytokinin biosynthetic genes *IPT3*, *IPT5*, and *IPT7* during FM termination (Figure 3a) [155]. In addition, ETT indirectly represses the expression of several *LOG* genes to fine-tune cytokinin levels [155]. In addition to controlling cytokinin levels, ETT controls the level of cytokinin signaling. ETT binds to the *AHK4* promoter to repress its expression (Figure 3a) [155]. AuxREs play key roles in the direct binding of ETT to *IPT7* and *AHK4* [155]. The role of auxin downstream of ETT adds multiple layers of complexity to the role of cytokinin in regulating FM termination to provide cells for subsequent gynoecium formation.

### 3.4. Proper Control of Auxin Homeostasis Terminates Meristematic Fate in the FM

During FM termination, AG also activates its direct target, *CRABS CLAW* (*CRC*), to enable FM determinacy via auxin (Figure 3b) [163,164,165,166]. AG binds to the *CRC* promoter and promotes its expression. *CRC* belongs to the YABBY family of transcription factors, which contain a zinc finger and a helix-loop-helix domain. *CRC* expression begins during stage 6 of flower development and localizes to the abaxial region of the developing carpels [157]. In addition to AG, CRC increases auxin levels through the direct activation of *YUC4* [167]. AG interacts with the SWI/SNF chromatin remodelers and evicts well-positioned nucleosomes (Figure 3c) [141,168,169,170]. The pioneer activity of AG, together with SWI/SNF chromatin remodelers, opens up the promoter region of *YUC4* from stage 3 of flower development (Figure 3c) [167]. Subsequently, CRC associates the *YUC4* promoter near the AG binding site (Figure 3c). The feed-forward activation of *YUC4* via the synergistic interaction between AG and CRC specifies primordium cell fate. Consistent with this finding, mutations in multiple *YUC* genes, including *YUC4*, induce FM indeterminacy. CRC also represses another direct target, *TORNADO2* (*TRN2*) (also known as *TETERASPANIN1* or *EKEKO***)**, which encodes a transmembrane protein of the tetraspanin family [171,172,173,174]. Tetraspanins interact with numerous partner proteins to control multiple cellular processes, such as cell adhesion, cell fusion, and intracellular membrane trafficking. This transcriptional repression of *TRN2* by CRC partially contributes to the creation of auxin maxima by interfering with auxin transport [175]. How TRN2 affects auxin accumulation remains to be addressed. These two direct CRC targets, *YUC4* and *TRN2*, act in parallel to repress *WUS* expression for FM determinacy and subsequent organ formation (Figure 3b) [171]. The sequential actions of transcription factors control auxin homeostasis in the FM for subsequent carpel formation.

### 3.5. Synergistic Interaction between Auxin and Cytokinin Promotes Gynoecium Formation

Although it is well known that auxin and cytokinin act antagonistically during plant growth and development, several studies on the molecular mechanism between these phytohormones have focused on their synergistic activity, which maximizes their effects during gynoecium formation [54,55,56]. Unlike the SAM, in carpels, the auxin and cytokinin accumulation patterns partially overlap after FM termination. SPATULA (SPT), a bHLH transcription factor, induces cytokinin accumulation in the carpel margin via an unknown mechanism [163,176,177,178]. SPT activates the expression of the auxin biosynthesis gene *TRYPTOPHAN AMINOTRANSFERASE OF ARABIDOPSIS1* (*TAA1*) and the auxin efflux carrier gene *PIN3* [177,179,180,181]. This expression change also promotes auxin accumulation in the apical region of the carpel. One remaining question that needs to be addressed is how one transcription factor mediates the activities of two antagonistic hormones in the same manner. The target specificity of SPT could be due to the presence of cofactors and/or the different affinities of SPT binding sites. Since most transcription factors that regulate gynoecium development are categorized into only four classes based on mRNA distribution [182], threshold regulation at the protein level could also be important. SPT forms multiple complexes with transcription factors, such as HEC and INDEHISCENT (IND) [163,176,183]. Proteomic-based studies of protein interactions might help reveal cofactors of SPT that function in response to auxin or cytokinin.

## 4. Concluding Remarks and Future Perspectives

The robust establishment of the SAM and FM is primarily controlled by conserved secreted peptides and master transcriptional regulators. In the past five years, plant stem cell researchers have identified various interactions between these key factors and phytohormones using a candidate gene approach or omics data. Phytohormones help determine when and where these factors function. The spatiotemporal fine-tuning of these factors by phytohormones appears to be regulated by complex multilayered networks of hormonal components, since hormones affect not only transcription, but also the epigenetic state, protein stability/protein–protein interactions, and metabolic rates of meristematic cells. However, in practice, most omics data are available only for bulk samples rather than single cells. Furthermore, technologies to integrate multilayer omics data are not yet well established. To elucidate these networks, multi-omics data at the single-cell level are needed. The integration of multilayered omics data via high-resolution readouts across hierarchies will provide new insights into phytohormone-mediated meristem activities.

## Figures and Tables

**Figure 1 ijms-20-04065-f001:**
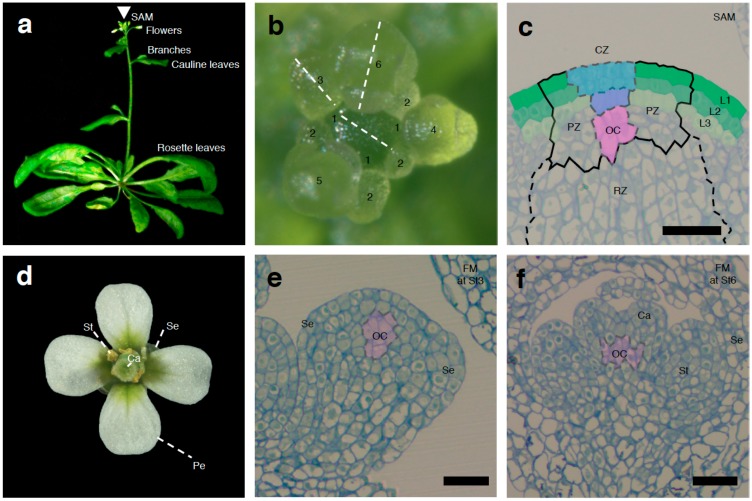
Morphology of the shoot apical meristem (SAM) and floral meristems (FM). (**a**) Side view of an Arabidopsis plant. Aerial tissues form from the SAM, which is located at the top of the plant (arrowhead). Thirty-day-old plants are shown. Lateral organs such as rosette leaves, cauline leaves, branches, and flowers form from the SAM. (**b**) Overhead view of the SAM. Numbers indicate floral stages [57]. The longitudinal sections shown in (**c**,**e**,**f**) were obtained by cutting across the SAM, stage 3 FM and stage 6 FM along the dashed lines, respectively. Thirty-day-old SAMs and FMs are shown. (**c**) Organization of the SAM showing functional zones and cell layers. The central zone (CZ) consists of stem cells (blue) and the organizing center (OC) (pink). PZ and RZ are the peripheral and rib zones, respectively. Epidermal (L1) and subepidermal (L2) layers are shown in green and light green, respectively. Scale bar = 20 µm. (**d**) Top view of an Arabidopsis flower at stage 13. A flower consists of four sepals, six stamens, four petals, and two carpels. A flower from a 30-day-old plant is shown. (**e**) Top view of the FM at stage 3. Scale bar = 20 µm. (**f**) Top view of the FM at stage 6. Organization of the FM showing functional zones and cell layers. The OC, which exhibits weak *WUS* expression only at stage 3 and early stage 6, is shown in pink. Scale bar = 20 µm.

**Figure 2 ijms-20-04065-f002:**
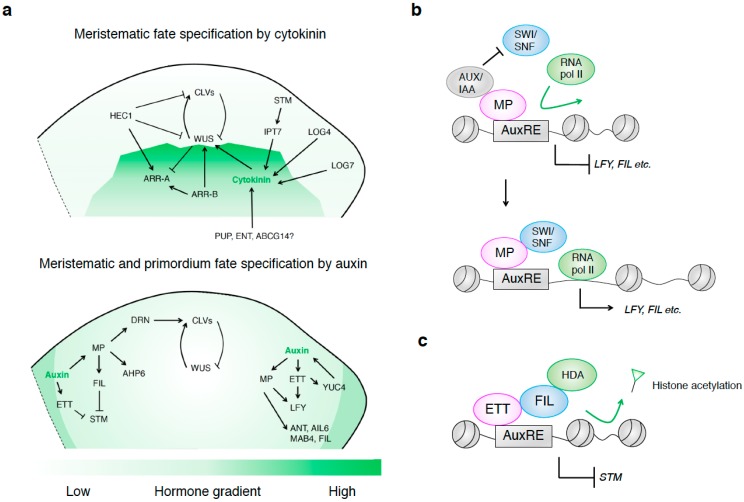
Hormonal control of the SAM to specify meristematic and primordium fate. (**a**) Top: Cytokinin-mediated SAM regulatory network for meristematic fate specification. Bottom: Auxin-mediated SAM regulatory network for meristematic and floral primordium fate specification. Green indicates hormonal gradients. The arrows represent the activation of gene expression, while the flat arrows represent its repression. (**b**) Chromatin state switch for primordium specification by the ARF5/MP complex in response to auxin. The arrows represent the activation of gene expression, while the flat arrows represent its repression. (**c**) Chromatin state switch for primordium specification by ARF3 and the FIL complex. The arrows represent the activation of gene expression, while the flat arrows represent its repression.

**Figure 3 ijms-20-04065-f003:**
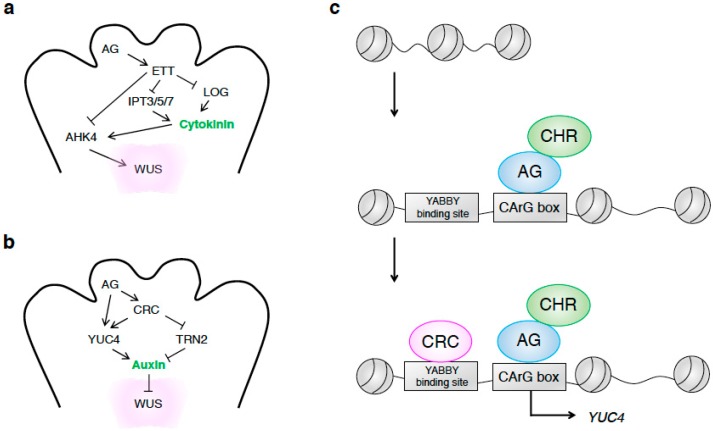
Hormonal control of the termination of floral meristematic fate. (**a**) Cytokinin-mediated meristematic fate termination in the FM at stage 6 of flower development. (**b**) Auxin-mediated primordium fate specification in the FM. The terminating OC is shown in pale pink. The arrows represent the activation of gene expression, while the flat arrows represent its repression. (**c**) Chromatin state switch for termination of floral meristematic fate by CRC and the AG complex. The arrow represents the activation of gene expression.

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
