# Peer review of "The Roles of Plant Hormones and Their Interactions with Regulatory Genes in Determining Meristem Activity"

_ijms, 2019, doi:10.3390/ijms20164065_

Round 1
Reviewer 1 Report
This is a nice review summarizing research about Plant hormones and how they regulate meristem development. Some minor changes should be considered before publication:
1) It is better to add more details or even a separate session talking about cross talks / interactions between these two hormones and how they work antagonistically and synergistically.
2) Although this paper focused on Arabidopsis, it is better to also talk about what research has been done on other plants like rice or maize.
3) Some references are missing. For example: Lines 86-87: "Although those two hormones are originally considered as antagonists, recent studies revealed their synergistic interaction as well." Please also check other places.
4) Proof-read the whole manuscript and correct some typos. For example: Line 254, "cytokinin regulates FM ... " Should capitalize the letter C. Please also check other places.
Author Response
Comments to Reviewer 1
General comment byReviewer 1
This is a nice review summarizing research about Plant hormones and how they regulate meristem development. Some minor changes should be considered before publication:
General response
We are grateful to Reviewer 1 for the favorable comments and feedback, which have helped us improve our paper. In particular, we should have included more information about the crosstalk between the two hormones (section 3) and the studies performed in other species (section 1). We have made all of the suggested revisions. We hope that our manuscript is now suitable for publication. Please seeour point-by-point responses below.
Request 1 by Reviewer 1
1) It is better to add more details or even a separate session talking about cross talks / interactions between these two hormones and how they work antagonistically and synergistically.
Response 1
In the revised version of our manuscript, we provide a separate section about crosstalk between auxin and cytokinin. Althoughhow these hormones function antagonistically and synergistically is not yet fully understood, we summarize the current knowledge in Section 3.5.
Request 2 by Reviewer 1
2) Although this paper focused on Arabidopsis, it is better to also talk about what research has been done on other plants like rice or maize.
Response 2
Thank you for pointing this out. We now discuss known conserved mechanisms in higher plants (CLV-WUS signaling components, LOG, and so on). We now cite more than 10 publications in rice, maize, and tomato.
Request 3 by Reviewer 1
3) Some references are missing. For example: Lines 86-87: "Although those two hormones are originally considered as antagonists, recent studies revealed their synergistic interaction as well." Please also check other places.
Response 3
We apologize for not citing several key references. More than 90 new references were added to the revised version of our manuscript (including Lines 86-87).
Request 4 by Reviewer 1
4) Proof-read the whole manuscript and correct some typos. For example: Line 254, "cytokinin regulates FM ... " Should capitalize the letter C. Please also check other places.
Response 4
Thank you for pointing this out. We have corrected the typo. Based on comments from the two other reviewers, many typos and errors were corrected (please see Minor request 4 by Reviewer 2 and Request 10-18 by Reviewer 3). Furthermore, the revised version of our review was edited by Plant Editors (https://planteditors.com/why-plant-editors/). We believe that the revised version of our review is presented clearly and accurately.
Reviewer 2 Report
Lee et al. provided a concise and focused review on the crosstalk between genetic and hormonal regulations in Arabidopsis aboveground meristems. The focus on the two developmentally important hormones, namely auxin and cytokinin, makes sense, and the comparison between an indeterminate (SAM) and determinate (FM) meristem is rather interesting: it highlights how similar regulations can be tuned to establish completely different developmental outputs. The authors should however make sure that several definitions are stringent and correct, or provide and discuss about different opinions if the authors wish to push forward alternatives (see below).
For example, in Lines 46-48 & 111, the central zone (CZ) is described to “consists of both stem cells and the organizing center (OC)”. But classically, CZ only consists of the stem cells (or more specifically CLV3 gene expression domain), while OC is not part of CZ and is identified by WUS gene expression. Nowadays it is accepted that CZ and OC partially overlap, and the authors may elaborate on it if wish to, but OC is not a subdomain of CZ. CZ + OC do form the “stem cell niche” though (depend on definition).
Also, in Fig. 1d, the proposed hormone gradients do not accurately reflect the current opinions. In short, CK gradient is at least not excluded from OC (left), and auxin gradient is also not as simplistic (right). The blue and pink color in Fig. 1d makes it difficult to paint hormone gradients in CZ and OC. The authors may consider removing these two colors and replacing them with green gradients, or keep all three colors but apply correct gradient to all of them to indicate hormone gradients across the whole meristem.
Other comments:
There’s a general void of in-text references to key claims (for example in Lines 48-58 and Lines 92-97, there are other sections that require more references), even though many appropriate references are present in the list. The authors should simply refer to them more frequently in text, mainly following key conclusions from previous works.
Both in the Abstract and Introduction, the word “environment” was pushed forward. Yet besides scarce mentioning of light, dark and nitrate, not much environment was discussed. Given the focus of this Special Issue, the authors can be more upfront with their discussion on the environmental influences, and may elaborate on epigenetics and developmental reproducibility / robustness to support the environmental discussion from different angles.
When describing (the rather complex) gene regulatory networks, the authors should refer back to their figures at least once per paragraph. In fact, there are no more figure references after Introduction…
The manuscript also has many minor conceptual issues, grammatical errors and typos. Here’s a non-exhaustive list:
Line 37: … top of the plant, (insert comma) is responsible for …
Lines 38 & 107: SAM does not give rise to cotyledons and hypocotyl, so not “all” aerial structures.
Lines 41-42: … anticlinal to the plain. Advice to define plain as “of tissue surface” or “of cell layer” or something else to be more specific, especially for L3 since it’s not planar. Also for L3, are there also circumferential and oblique divisions?
Line 57: … heterodimerize “with” (insert) the “receptor-like cytoplasmic” (swap) kinase CORYNE …
Line 90: The heading is a bit clunky… What about “Hormonal control of meristemic and primordial fates in the SAM”?
Line 92: … which triggers cell division, (add comma) was first …
Fig 2d: Since arrows were stated as gene expression regulations, shouldn’t the genes in Fig. 2d be italic?
Line 128: … contribute to “producing” cytokinin, or contribute to cytokinin “production”.
Line 132: … OXIDASE5 (missing “I”).
Line 133: … are enzymes that “catalyze” (delete “s”) …
Line 134: … in “the” dark, or in “darkness”.
Line 139: … mediated “both by” multiple …
Line 140: “hence” needed?
Line 143: It has “been” shown, or it “was” shown …
Lines 165-167: Sentences are confusing.
Line 183: … a “classic” role …
Line 289: … in Figure “1b” and 2b …
Line 297: “Highly dynamic yet stable”? FM is not indeterminate so is not stable, right? Do you mean “robust pattern”, “robust establishment” or something else in that line?
Etc.
Author Response
Comments to Reviewer 2
General comment byReviewer 2
Lee et al. provided a concise and focused review on the crosstalk between genetic and hormonal regulations in Arabidopsis aboveground meristems. The focus on the two developmentally important hormones, namely auxin and cytokinin, makes sense, and the comparison between an indeterminate (SAM) and determinate (FM) meristem is rather interesting: it highlights how similar regulations can be tuned to establish completely different developmental outputs. The authors should however make sure that several definitions are stringent and correct, or provide and discuss about different opinions if the authors wish to push forward alternatives (see below).
General response
We are grateful to Reviewer 2 for the critical feedback, which has helped us improve our paper. In particular, we should have clarified all definitions (Requests 1 and 2). Therefore, we have modified both the text and figures, as the reviewer suggested. Furthermore, we have conducted all of the suggested revisions. We hope that our manuscript will now be deemed suitable for publication. Please seeour point-by-point responses below.
Request 1 by Reviewer 2
For example, in Lines 46-48 & 111, the central zone (CZ) is described to “consists of both stem cells and the organizing center (OC)”. But classically, CZ only consists of the stem cells (or more specifically CLV3 gene expression domain), while OC is not part of CZ and is identified by WUS gene expression. Nowadays it is accepted that CZ and OC partially overlap, and the authors may elaborate on it if wish to, but OC is not a subdomain of CZ. CZ + OC do form the “stem cell niche” though (depend on definition).
Response 1
We fully agree that we should have clarified the definitions of each zone. In the revised version of our review, new sentences were included describing the central zone and organizing center. In addition, we explain that CZ and OC partially overlap and form the stem cell niche. Finally, the overlapping region is shown in the revised version of Fig. 1.
Request 2 by Reviewer 2
Also, in Fig. 1d, the proposed hormone gradients do not accurately reflect the current opinions. In short, CK gradient is at least not excluded from OC (left), and auxin gradient is also not as simplistic (right). The blue and pink color in Fig. 1d makes it difficult to paint hormone gradients in CZ and OC. The authors may consider removing these two colors and replacing them with green gradients, or keep all three colors but apply correct gradient to all of them to indicate hormone gradients across the whole meristem.
Response 2
We fully agree that the original version of Fig. 1 did not accurately reflect hormone gradients. Since the purpose of providing Fig. 2a is to show hormonal gradients, we simply removed the blue and pink colors and corrected the diagram of the gradients.
Minor request 1 by Reviewer 2
Other comments: There’s a general void of in-text references to key claims (for example in Lines 48-58 and Lines 92-97, there are other sections that require more references), even though many appropriate references are present in the list. The authors should simply refer to them more frequently in text, mainly following key conclusions from previous works.
Minor response 1
We apologize for not citing several key references. In the revised version of our review,more than 90 new references were added to the appropriate sentences (including Lines 48-58 and Lines 92-97).
Minor request 2 by Reviewer 2
Both in the Abstract and Introduction, the word “environment” was pushed forward. Yet besides scarce mentioning of light, dark and nitrate, not much environment was discussed. Given the focus of this Special Issue, the authors can be more upfront with their discussion on the environmental influences, and may elaborate on epigenetics and developmental reproducibility / robustness to support the environmental discussion from different angles.
Minor response 2
Reviewer 3 (request 4) indicated that "While its true that the activity of the meristem would be modulated by the environment (for instance temperature and light conditions would decide about the switch to bolting), the continuous meristematic activity is rather a basic feature of plant development." Thus, we toned down the discussion of environmental effects on the regulation of meristem activity in the revised version of our review. However, we included several sentences in the Discussion about the effects of the environment on meristem activity. Furthermore, we now describe more epigenetic effects in the text and figures.
Minor request 3 by Reviewer 2
When describing (the rather complex) gene regulatory networks, the authors should refer back to their figures at least once per paragraph. In fact, there are no more figure references after Introduction…
Minor response 3
We agree with this point. To address this issue, 1) we reorganized the figures and provided a figure describing the general structures of the SAM and FM, as described in the Introduction; and 2) in each section, a figure is provided and referred to properly.
Minor request 4 by Reviewer 2
The manuscript also has many minor conceptual issues, grammatical errors and typos. Here’s a non-exhaustive list:
Line 37: … top of the plant, (insert comma) is responsible for …
Lines 38 & 107: SAM does not give rise to cotyledons and hypocotyl, so not “all” aerial structures.
Lines 41-42: … anticlinal to the plain. Advice to define plain as “of tissue surface” or “of cell layer” or something else to be more specific, especially for L3 since it’s not planar. Also for L3, are there also circumferential and oblique divisions?
Line 57: … heterodimerize “with” (insert) the “receptor-like cytoplasmic” (swap) kinase CORYNE …
Line 90: The heading is a bit clunky… What about “Hormonal control of meristemic and primordial fates in the SAM”?
Line 92: … which triggers cell division, (add comma) was first …
Fig 2d: Since arrows were stated as gene expression regulations, shouldn’t the genes in Fig. 2d be italic?
Line 128: … contribute to “producing” cytokinin, or contribute to cytokinin “production”.
Line 132: … OXIDASE5 (missing “I”).
Line 133: … are enzymes that “catalyze” (delete “s”) …
Line 134: … in “the” dark, or in “darkness”.
Line 139: … mediated “both by” multiple …
Line 140: “hence” needed?
Line 143: It has “been” shown, or it “was” shown …
Lines 165-167: Sentences are confusing.
Line 183: … a “classic” role …
Line 289: … in Figure “1b” and 2b …
Line 297: “Highly dynamic yet stable”? FM is not indeterminate so is not stable, right? Do you mean “robust pattern”, “robust establishment” or something else in that line?
Minor response 4
Thank you for reading the original version of our review carefully. We corrected
all mistakes and typos based on comments from all three reviewers. Furthermore, the revised version of our review was edited by Plant Editors (https://planteditors.com/why-plant-editors/). We believe that the revised version of our review is presented clearly and accurately.
Reviewer 3 Report
The review is overall quite well-written and integrates a great deal of information. That said, there are multiple points where it could be still improved to increase the value of this manuscript and its usability as an introduction to the topic. In other places I had a bit more serious concerns, such as that regarding the basic interpretation of the meaning of continuous meristematic activity in plants; in some parts of the text, references appear to be missing, which for a review is also a serious flaw.
1) Introduction
It feels that the Introduction lacks some basic explanation of an apical meristem function. A description of how primordia gradually arise from the meristem, in phyllotactic patterns, to continuously produce organs while the meristem itself renews, is lacking somewhere in the beginning. Perhaps similarly a picture in Fig. 1 showing the same concept would help. Right now, the concept of organ production comes somewhat without explanation in lines 44-45.
The Figure also does not include the CLV1,2 receptors. Perhaps the figure could go into more specific details, with drawing of the receptors, and the ligands, instead of just arrows indicating activation and repression?
In the ontogenetic context, it could be described in the text what are the products of the Arabidopsis SAM over time: first, rosette leaves, then, cauline leaves together with lateral buds, and finally, flowers. Such broad overview of the meaning of the meristem would help, as now, there is an impression that the SAM first produces undefined lateral organs, and then, floral meristems.
Also, I would argue with the interpretation found in both the Abstract and Intoduction, that the continuous post-embryonic meristematic activity in plants is intended for adaptation to the plant environment. While its true that the activity of the meristem would be modulated by the environment (for instance temperature and light conditions would decide about the switch to bolting), the continuous meristematic activity is rather a basic feature of plant development, without which the plant would forever remain a tiny seedling without more than 1 or 2 cotyledons, a single primary root, and definitely without any reproductive organs. The need to develop all these organs has nothing to do with adverse environmental conditions or lack thereof.
When discussing the determinancy of FM compared to the indeterminacy of the SAM, it could be explained that SAMs can also be determinate in many species or varieties within species of plants.
Regarding the introduction on floral meristems, perhaps some words about the ABC model would be fitting. Also, in lines 69-70, it could be explained which flower morphology phenotypes arise from early or late FM termination, together with references.
The figure similarly to the text could explain a bit more about the architecture of the flower and how this architecture arises in the meristem. The figure does not really explain how a floral meristem looks, what are its component organ primordia, etc. It's hard to understand anything in detail from panel b. Also, the text in Fig 2d is a bit too small, similarly to 1d.
40 It would not hurt to introduce the classical tunica and corpus concepts next to the modern distinction of L1, L2 and L3.
41-42 "anticlinally", "periclinally"; plain of...? perhaps: plain of the tissue layer
2) Cytokinin and auxin chapters
93-94 Please add references
173-178 Please add references
Please make sure that all information throughout the manuscript contains necessary references.
99-101 When reading these lines, I had the impression that transport from the roots is a major mode of cytokinin action in the SAM, but just in the next sentence, as well as from reading the following paragraphs, it is clear that it is not really so, as local controls by biosynthesis and other play a major role.
135 nutritational status?
144 has been shown
158, 247 "allow" and "allows"
185 It may be added that interestingly, the earlier product of SAM, rosette leaves, are generally present.
196 Please include a few more specific words about the auxin receptor complex including SCF^TIR1/AFB. Leyser 2018 can be a useful reference.
It would be great if each chapter ends with a short and simple summary of the main modes of action of the hormone. Overall, the text is quite complex, and perhaps to some degree, it feels like a listing of facts, rather than an extraction of the essence by integrating information from various sources together. Perhaps the Authors could try to improve a bit on the streamlining of the text.
4) Hormonal control of the FM
It feels that this chapter contains some repetitions. For instance the same facts about IPT3,5,7 are written twice in lines 248 and 260. Perhaps the Authors can have another look at this whole chapter to make sure if it can't be a bit straightened.
278 "a lot" is a colloquial expression
Author Response
Comments to Reviewer 3
General comment byReviewer 3
The review is overall quite well-written and integrates a great deal of information. That said, there are multiple points where it could be still improved to increase the value of this manuscript and its usability as an introduction to the topic. In other places I had a bit more serious concerns, such as that regarding the basic interpretation of the meaning of continuous meristematic activity in plants; in some parts of the text, references appear to be missing, which for a review is also a serious flaw.
General response
We are grateful to Reviewer 1 for the critical comments, which have helped us improve our paper. In particular, we should have described meristem activity and morphology more clearly. Based on your comments, we have modified our text and figures and now cite important references. We hope that our manuscript is now suitable for publication. Please seeour point-by-point responses below.
Request 1 by Reviewer 3
It feels that the Introduction lacks some basic explanation of an apical meristem function. A description of how primordia gradually arise from the meristem, in phyllotactic patterns, to continuously produce organs while the meristem itself renews, is lacking somewhere in the beginning. Perhaps similarly a picture in Fig. 1 showing the same concept would help. Right now, the concept of organ production comes somewhat without explanation in lines 44-45.
Response 1
We fully agree that we should have described the general function of the apical meristem. In the revised version of our paper, we included a description of how primordia gradually arise from the meristem, in phyllotactic patterns, to continuously produce organs while the meristem itself is renewed. Also, we highlighted phyllotactic patterning in the new version of Fig. 1b.
Request 2 by Reviewer 3
The Figure also does not include the CLV1,2 receptors. Perhaps the figure could go into more specific details, with drawing of the receptors, and the ligands, instead of just arrows indicating activation and repression?
Response 2
We now show CLV1 and CLV2 in the modified version of the figure. We also included new figures describing the interactions between hormonal components and transcriptional/epigenetic regulation during SAM and FM development.
Request 3 by Reviewer 3
In the ontogenetic context, it could be described in the text what are the products of the Arabidopsis SAM over time: first, rosette leaves, then, cauline leaves together with lateral buds, and finally, flowers. Such broad overview of the meaning of the meristem would help, as now, there is an impression that the SAM first produces undefined lateral organs, and then, floral meristems.
Response 3
As you pointed out, to provide a broad overview of meristem action, it was important to make the paper more broadly accessible to readers of high-impact journals such asInternational Journal of Molecular Science.In the revised version of the review, we describe the products of the SAM during the vegetative and reproductive phases in the Introduction.
Request 4 by Reviewer 3
Also, I would argue with the interpretation found in both the Abstract and Introduction, that the continuous post-embryonic meristematic activity in plants is intended for adaptation to the plant environment. While its true that the activity of the meristem would be modulated by the environment (for instance temperature and light conditions would decide about the switch to bolting), the continuous meristematic activity is rather a basic feature of plant development, without which the plant would forever remain a tiny seedling without more than 1 or 2 cotyledons, a single primary root, and definitely without any reproductive organs. The need to develop all these organs has nothing to do with adverse environmental conditions or lack thereof.
Response 4
We fully agree with this comment. To tone down our discussion of this point, we removed the statements about environmental responses from the Abstract and Introduction in the revised version of our review. At the same time, recent studies have suggested that meristem size and hormone levels might be modulated by environmental conditions. Given the focus of this Special Issue, we left these references in the manuscript and provided more discussion of environmental influences.
Request 5 by Reviewer 3
When discussing the determinancy of FM compared to the indeterminacy of the SAM, it could be explained that SAMs can also be determinate in many species or varieties within species of plants.
Response 5
Thank you for pointing this out. In the revised manuscript, we explain that SAMs can also be determinate, depending on the species.
Request 6 by Reviewer 3
Regarding the introduction on floral meristems, perhaps some words about the ABC model would be fitting. Also, in lines 69-70, it could be explained which flower morphology phenotypes arise from early or late FM termination, together with references.
Response 6
In the revised version of our review, we briefly explain the role of the ABC gene in floral organ specification. Also, we discuss when and where floral organs arise from the FM in the revised version of the Introduction.
Request 7 by Reviewer 3
The figure similarly to the text could explain a bit more about the architecture of the flower and how this architecture arises in the meristem. The figure does not really explain how a floral meristem looks, what are its component organ primordia, etc. It's hard to understand anything in detail from panel b. Also, the text in Fig 2d is a bit too small, similarly to 1d.
Response 7
In the revised version of our review, we show the architectures of the shoot apical meristem, the floral meristems and flowers, and describe how floral organs form, including a description ofthe ABC model. We also enlarged the figures showing gene regulatory networks.
Request 8 by Reviewer 3
40 It would not hurt to introduce the classical tunica and corpus concepts next to the modern distinction of L1, L2 and L3.
Response 8
We describe the classical tunica and corpus concepts in the revised version of the Introduction.
Request 9 by Reviewer 3
41-42 "anticlinally", "periclinally"; plain of...? perhaps: plain of the tissue layer
Response 9
We explain the plane of the tissue layer in the revised version of the manuscript. Thank you for this comment.
Request 10 by Reviewer 3
93-94 Please add references
Response 10
A few references were added to former Line 93-94.
Request 11 by Reviewer 3
173-178 Please add references. Please make sure that all information throughout the manuscript contains necessary references.
Response 11
The necessary information was added to former Line 173-178. Also, we carefully checked the references and added more than 90 new references to the revised version of the manuscript.
Request 12 by Reviewer 3
99-101 When reading these lines, I had the impression that transport from the roots is a major mode of cytokinin action in the SAM, but just in the next sentence, as well as from reading the following paragraphs, it is clear that it is not really so, as local controls by biosynthesis and other play a major role.
Response 12
We apologize for the confusion. In the revised version of our review, we emphasize the importance of not only cytokinin transport, but also cytokinin biosynthesis and signaling.
Request 13 by Reviewer 3
135 nutritational status?
144 has been shown
158, 247 "allow" and "allows"
Response 13
Thank you for pointing out these typos and mistakes. We have corrected all of them.
Request 14 by Reviewer 3
185 It may be added that interestingly, the earlier product of SAM, rosette leaves, are generally present.
Response 14
In the revised version of our review, we describe thempphenotype during the vegetative phase.
Request 15 by Reviewer 3
196 Please include a few more specific words about the auxin receptor complex including SCF^TIR1/AFB. Leyser 2018 can be a useful reference.
Response 15
Thank you for introducing us to this useful review paper. We described the auxin receptors in the revised manuscript and added references.
Request 16 by Reviewer 3
It would be great if each chapter ends with a short and simple summary of the main modes of action of the hormone. Overall, the text is quite complex, and perhaps to some degree, it feels like a listing of facts, rather than an extraction of the essence by integrating information from various sources together. Perhaps the Authors could try to improve a bit on the streamlining of the text.
Response 16
We agree with this point. In the revised version of our review, we have included a short and simple summary.
Request 17 by Reviewer 3
It feels that this chapter contains some repetitions. For instance the same facts about IPT3,5,7 are written twice in lines 248 and 260. Perhaps the Authors can have another look at this whole chapter to make sure if it can't be a bit straightened.
Response 17
We apologize for the confusion. ETT regulates IPT3, IPT5, and IPT7not only during stage 3, but also during stage 6. To avoid this confusion, we inserted subheadings.
Request 18 by Reviewer 3
278 "a lot" is a colloquial expression
Response 18
The phrase "a lot" was changed to "multiple" in the revised version of our review.
Round 2
Reviewer 3 Report
I think the manuscript is significantly improved and I would recommend its publication. I only have two concerns, please see below.
Between 128 and 129: A section header "2.1" is missing.
276 A very surprising sentence: "The organogenic program that terminates the meristem is conserved between plants and animals.". It is really unclear what this sentence refers to, which components described in this paragraph are conserved. I can hardly believe there could be a whole conserved organogenic program between plants and animals, since multicellular development of plants and animals evolved separately and animals don't have plant-like meristems?